

# The source, quantity, and spatial distribution of interfacial water during glide-snow avalanche release: experimental evidence from field monitoring

Amelie Fees[1], Michael Lombardo[1], Alec van Herwijnen[1], Peter Lehmann[2], and Jürg Schweizer[1]

[1]WSL Institute for Snow and Avalanche Research SLF, Davos, Switzerland
[2]Physics of Soils and Terrestrial Ecosystems, ETH Zürich, Switzerland

**Correspondence:** Amelie Fees (amelie.fees@slf.ch)

**Abstract.**

Glide-snow avalanches release at the soil-snow interface due to a loss friction which is suspected to be linked to interfacial water. To date, the formation and distribution of the interfacial water are not well understood, and glide-snow avalanches are considered unpredictable. We investigated the source, quantity, and spatial distribution of interfacial water before and during avalanche release through spatio-temporal field monitoring. The measurement setup consists of a sensor grid covering a slope with frequent glide-snow avalanche activity. The 24 grid sensors measured the soil temperature and liquid water content (LWC) throughout the seasons 2021/22 to 2023/24. Snow/interfacial temperature and LWC were monitored locally with a vertical sensor profile ranging from the soil into the snow. Seven glide-snow avalanches released on the sensor grid and their investigation showed: (i) interfacial water originated from geothermal heat, rain, and meltwater percolation, (ii) the quantity of snow LWC was lower for glide-snow avalanches that released in early winter than in spring, (iii) soil temperatures below the release area were higher than in the remaining slope if interfacial water originated from geothermal heat (iv) if interfacial water originated from rain/melt we observed (locally) higher soil LWC below the release area and (v) for four avalanches the spatial variability of soil LWC across the slope reached a local minimum at the time of avalanche release. In the future, with continued monitoring, the spatio-temporal investigation of the soil will help to quantify the drivers for glide-snow avalanche release at the slope scale. This will contribute to improved glide-snow avalanche forecasting and mitigation.

## 1 Introduction

Glide-snow avalanches release at the soil-snow interface and endanger infrastructure in mountain regions (e.g. Clarke and McClung, 1999; Mitterer and Schweizer, 2012a). The snow gliding process has been reported on since the early 20th century (Frankhauser, 1918; Haefeli, 1939). Since then, many phenomenological observations have led to our current understanding that snow gliding is favored by (i) a smooth ground surface, (ii) sufficiently steep slopes above 15° (typically above 28° for glide-snow avalanches) and (iii) water at the soil-snow interface that causes a reduction in basal friction (Ancey and Bain, 2015; in der Gand and Zupančič, 1966). Glide-snow avalanches have been observed both in early winter and in spring, which led to the classification of "cold" and "warm" glide-snow avalanches based on prevailing air and snow temperature (Clarke



and McClung, 1999; Dreier et al., 2016). It is generally assumed that the source of interfacial water differs between "cold"
and "warm" events depending on the interplay of meteorological, snowpack, and soil parameters. This motivated a similar
but more process-based separation of glide-snow avalanches into "interface" and "surface" events (Fees et al., 2023). Surface
events (surface-generated interfacial water) are avalanches where the water at the soil-snow interface originated from the snow
surface through percolation of meltwater or rain (Lackinger, 1987; Clarke and McClung, 1999). Interface events (interface-
generated interfacial water) are avalanches where the liquid water was formed at the soil-snow interface. Possible sources of
water for interface events include geothermal melting of the lowermost snow layer (McClung, 1987; Newesely et al., 2000;
Höller, 2001) or capillary suction of water from the soil into the snow due the hydraulic soil and snow properties (Mitterer and
Schweizer, 2012a; Lombardo et al., 2024).

   While there have been numerous phenomenological observations of snow-gliding and glide-snow avalanche release (e.g.
McClung et al., 1994; Reardon et al., 2006; Höller, 2001), we still lack process understanding which results in limited forecast-
ing capabilities (Simenhois and Birkeland, 2010; Jones, 2004) and hampers mitigation measures (Sharaf et al., 2008; Jones,
2004). The review papers on snow gliding and glide-snow avalanches (Ancey and Bain, 2015; Höller, 2014; Jones, 2004) agree
that "the crux of the problem is the proper determination of what happens at the interface between the snowpack and ground"
(Ancey and Bain, 2015). The connection of soil conditions, basal friction and avalanche activity "is primarily based on obser-
vations and not yet confirmed by relevant investigations" (Höller, 2014). As a result, the number of in-situ observations of soil
liquid water content (LWC), soil temperature (Ceaglio et al., 2017; Maggioni et al., 2019; Mitterer and Schweizer, 2012b) and
snow LWC (Fromm et al., 2018) have increased recently. Findings included increasing glide rates with increasing soil LWC
(Ceaglio et al., 2017) and a significant influence of soil LWC and soil temperature on snow gliding (Fromm et al., 2018). The
available in-situ soil and snow measurements (e.g. temperature, LWC) were recorded as single point measurements. While
the temporal resolution of these measurements was high, the sensors were rarely located below a glide-crack or avalanche
(Fromm et al., 2018; Ceaglio et al., 2017). A recent approach to modelling the distribution of glide-snow avalanche release
areas (Fees et al., 2024) suggested that the spatial variability of the basal friction is important for glide-snow avalanche release.
We suspect that the basal friction is linked to the presence of interfacial water, but to the best of our knowledge, the influence
of spatio-temporal soil/snow LWC on snow-gliding has not been investigated in the field.

   To investigate the source, quantity, and spatial distribution of the interfacial water before and during avalanche release we
installed a sensor grid for spatio-temporal measurements within a slope with frequent glide-snow avalanche activity. The 24
grid sensors measured the soil temperature and LWC across the slope. Temperature and LWC at the interface and in the
snow were monitored locally on the slope with a vertical sensor profile ranging from the soil into the snow. During seasons
2021/22 to 2023/24 a total of seven glide-snow avalanches released on the sensor grid. These events provide the basis for
our phenomenological investigation into the source, quantity, and spatial distribution of interfacial water before and during
avalanche release.



## 2   Field site

The study area is located on the mostly southeast-facing hillslope of the Salezer Horn called Dorfberg (1650 to 2100 m a.s.l., Davos, Switzerland, Figure 1). Meteorological data are recorded at four automated weather stations (AWS) in close proximity ranging in elevation from Davos to Weissfluhjoch (1563 - 2536 m a.s.l., Table 1). The average sum of new snow height is 4 m in Davos and 7 m at Weissfluhjoch (16-year average for seasons 2009 to 2024, Figure 2, season 2009 refers to winter season 2008/09). The glide-snow avalanche activity on Dorfberg was monitored with time-lapse photography since season 2009 and extracted using a semi-automated pixel detection algorithm which was introduced in Fees et al. (2023). In addition, we simulated the snowpack at eleven representative virtual stations on Dorfberg using the meteorological data from the AWS of Weissfluhjoch and Klosters-Madrisa as input and of Dorfberg (Table 1) as validation (Fees et al., 2023). The simulation was initiated without a soil and with the bucket-approach for meltwater percolation (for SNOWPACK setup and validation see Fees et al. (2023)). The SNOWPACK simulations were used to classify the avalanches in surface and interface events. Surface events were defined as events where the water at the soil-snow interface originated from the snow surface through percolation of meltwater or rain. In interface events the liquid water forms at the soil-snow interface which was defined through a lack of simulated meltwater formation (Fees et al., 2023).

**Table 1.** Weather stations in proximity to Dorfberg.

| Location | Elevation (m a.s.l.) | Distance to Dorfberg | IMIS station ID |
|---|---|---|---|
| Weissfluhjoch | 2536 | ∼2 km northwest | WFJ2 |
| Klosters Madrisa | 2147 | ∼10 km northeast | KLO2 |
| Davos, SLF | 1563 | ∼1 km southeast | SLF2 |
| Dorfberg | 2140 | | not IMIS |

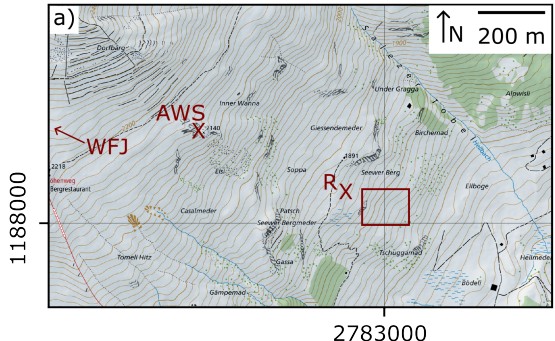
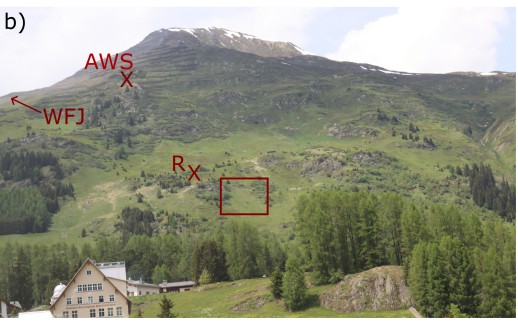

**Figure 1.** a) Map and b) picture of Dorfberg indicating the location of the weather station (AWS), the reference location (R), the Seewer Berg slope with the spatio-temporal monitoring setup (square), and the direction towards the Weissfluhjoch measurement site (WFJ). Map: Federal Office of Topography, CH1903+.





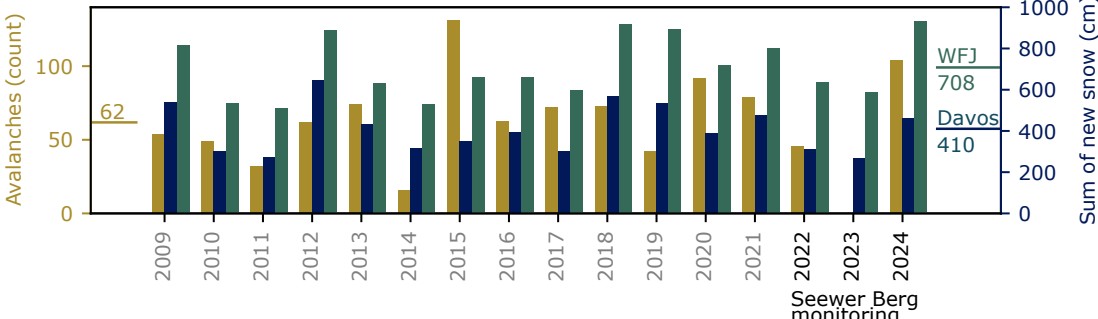

**Figure 2.** Observed number of glide-snow avalanches on Dorfberg and sum of new snow height (November to April) at Weissfluhjoch (2536 m a.s.l.) and in Davos (1563 m a.s.l.) for seasons 2009 to 2024. The average is indicated with the corresponding horizontal line. The Seewer Berg was monitored from season 2022 to 2024 which includes season 2023 when no glide-snow avalanche was recorded on Dorfberg.

The slope we monitored on Dorfberg is called Seewer Berg. This southeast-facing slope (46.8183° N, 9.8367° E; 1765 to 1818 m a.s.l; slope (31±5)°; Figure 1) experiences frequent glide-snow avalanche activity which is well documented through time-lapse photographs (Fees et al., 2023). The slope is mostly covered in long grass (Figure 3d) with interspersed shrubs and small rocky areas. There is no superficial water source within the slope, making it a suitable choice to investigate the source of interfacial water while excluding influences from groundwater sources. Approximately 100 m northwest of the Seewer Berg

slope is a shallow slope ($\approx 20°$) that was used as a reference site. This slope is protected from glide-snow avalanches and it was used for recording (bi)weekly manual snow profiles (Figure 3). Two soil profiles, taken in the Seewer Berg slope (location of vertical sensor profiles, Figure 3a) and in the reference location, indicated that the upper 30 cm of the soil is a sandy loam with relatively low densities ranging between 390 kg m$^{-3}$ and 890 kg m$^{-3}$ (Soil Survey Staff, 1999).

## 3   Spatio-temporal monitoring setup

From season 2022 to 2024, we continuously monitored the soil and snow conditions at the Seewer Berg slope using a total of 44 sensors across the entire slope (Table 2). The time interval between measurements was 15 minutes for all sensors and all liquid water content (LWC) measurements were capacity based.

**Soil:** The soil was monitored in 24 locations across the entire slope using a grid of combined soil LWC and temperature sensors (TEROS11, Meter Group). The sensors were installed at a soil depth of -5 cm. At this depth the sensor's measurement

volume (1010 cm$^3$, Meter Group (2024)) is covered by soil but is positioned as close to the soil-snow interface as possible. The grid spacing between sensors was approximately 8 m by 8 m (Figure 3a) and the maximum distance between two sensors is 52 m.

**Interface:** The soil-snow interface was monitored with a vertical profile of sensors ranging from a soil depth of -20 cm to a snow height of 25 cm (Figure 3b). The location of the vertical profile was in a common glide-snow avalanche release path



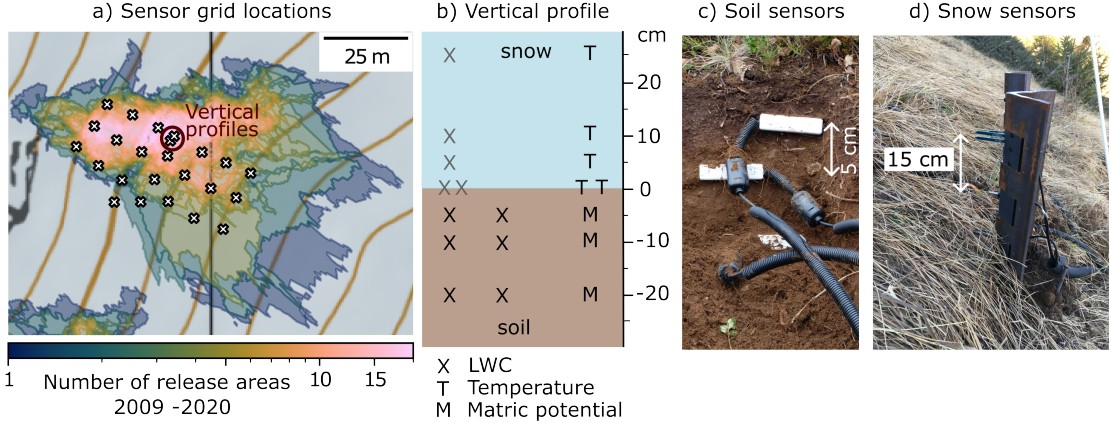

**Figure 3.** a) The Seewer Berg slope with the location of the soil LWC sensors (X). The maroon circle indicates the location of the vertical profiles across the soil-snow interface. b) Schematic of the vertical soil-snow profile. The soil LWC sensors and matric potential sensors have an integrated temperature sensor. c) Picture of a vertical soil LWC profile d) Picture of the wedges for snow LWC and temperature monitoring. Map: Federal Office of Topography, north up, contour line spacing: 10 m.

(Figure 3a) which was selected based on 13 years (2009-2021) of glide-snow avalanche activity extracted from time-lapse photographs (Fees et al., 2023). The vertical profiles included the following sensors: (i) in the soil (depths: -5 cm, -10 cm, -20 cm): two LWC/temperature sensors (TEROS 11) and one matric potential sensor at each depth (Tensiomark, ecoTech Umswelt-Messsysteme GmbH, Figure 3b,c). The matric potential sensors were installed in summer 2022. (ii) at the soil-snow interface (depth: 0 cm): two LWC sensors (EC5, Meter Group) and two temperature sensors (T107, Campbell Scientific) and

(iii) in the snow (heights: 5 cm, 10 cm and 20 cm): one LWC (EC5) and one temperature (T107) sensor at each height. The sensors in the snow were mounted on two vertical iron wedges (Figure 3d) with 3D-printed inlets to prevent metal from the wedges within the sensor's measurement volume. We used the EC5 sensors for measurements in the snow and at the interface due to their smaller measurement volume (240 cm$^3$) which allows for more localized measurements.

**Snow:** The snowpack was observed through (bi)weekly manual snow profiles at the reference site (Figure 1). Snow stratig-

raphy, including grain size and grain type, were recorded using a magnifying glass and grid (Fierz et al., 2009). Density and relative permittivity were measured every 5 to 10 cm to derive LWC (Denoth, 1994). Snow density was obtained from the average of two measurements per height with a cylindrical density cutter (100 cm$^3$). The relative permittivity was determined with a capacitive probe (Denoth, 1994). In case of a glide-snow avalanche in the Seewer Berg slope, an additional manual snow profile was recorded close to the fracture line, typically within one or two days after release.



**Table 2.** Overview of measured parameters, their location, and method or sensor used.

| | Parameter | Installation depth | Nr sensors | Time interval | Unit | Location | Method/Sensor |
|---|---|---|---|---|---|---|---|
| Soil | Liquid water content | (-5 cm, -10 cm, -20 cm) | (20, 2, 2) | 15 min | $m^3 m^{-3}$ | Seewer Berg | TEROS11 sensors |
| | Temperature | (-5 cm, -10 cm, -20 cm) | (24, 2, 2) | 15 min | °C | Seewer Berg | TEROS11 sensors |
| | Matric potential | (-5 cm, -10 cm, -20 cm) | (1,1,1) | 15 min | hPa | Seewer Berg | Tensiomark sensors |
| Soil-snow interface | Liquid water content | 0 cm | 2 | 15 min | a.u. | Seewer Berg | EC5 sensors |
| | Temperature | 0 cm | 2 | 15 min | °C | Seewer Berg | T107 sensors |
| Snow cover | Snow height | | | 10 min | cm | Reference site | SNOWPACK |
| | Snow height | | | (bi)weekly | cm | Reference site | manual snow profile |
| | Snow LWC | (5 cm, 10 cm, 25 cm) | (1,1,1) | 15 min | % | Seewer Berg | EC5 sensors |
| | Snow temperature | (5 cm, 10 cm, 25 cm) | (1,1,1) | 15 min | % | Seewer Berg | T107 sensors |
| | Snow LWC | $\Delta h = 5$ or 10 cm | | (bi)weekly | % | Reference site | manual snow profile |
| | Snow LWC (Avalanche) | $\Delta h = 5$ or 10 cm | | irregular | % | Seewer Berg | manual snow profile |
| | Snow stratigraphy | | | (bi)weekly | | Reference site | manual snow profile |
| | Snow stratigraphy (Avalanche) | | | irregular | | Seewer Berg | manual snow profile |
| | Surface-/interface classification | | | daily | | Reference site | SNOWPACK (Fees et al., 2023) |
| Meteorological | Air temperature | | | 10 min | °C | Reference site | SNOWPACK (MeteoIO) |
| Other | Avalanche activity | | | daily | | Dorfberg | time-lapse photography |
| | Avalanche release time | | | 5 min | | Seewer Berg | time-lapse photography |
| | continuous/patchy snow cover | | | 4 hours | | Seewer Berg | time-lapse photography |

## 4 Data processing

### 4.1 Measurements

**Soil:** We evaluated the reliability of the sensors using data from summer rainfalls. Four soil LWC sensors (locations indicated in Figure 4) responded slowly to the infiltrating water compared to the rest of the sensors. These sensors may not have been fully connected to the soil matrix and we excluded them from soil LWC observations during the winter seasons. When an avalanche released over the sensor grid we separated the grid sensors into sensors in (below) the release area and sensors outside the release area (Figure 4). To identify the sensor in the release area, we extracted the release area from time-lapse photographs. However, we previously observed that extracting the release area from the time-lapse photographs tends to underestimate the release area, especially for small avalanches (Fees et al., 2023). We manually added sensors when we observed additional snow free sensors in the field while recording the manual snow profiles in the release area.

**Interface:** The interface LWC sensors were installed within the vegetation. The measurement volume of these sensors extended from the soil, across the (vegetation) interface, and into the snow. As a result, they measured a combination of soil, vegetation, and snow LWC, making a sensor calibration for quantitative analysis difficult. We used the raw values (arbitrary units: a.u.) to investigate relative changes.

**Snow:** To monitor the LWC within the snow, we deployed three sensors on metal wedges (Figure 3d). When the snow cover did not sufficiently cover the wedges, preferential melt occurred around the wedges and the sensors measured the permittivity of air and meltwater instead of snow. To exclude these measurements, time periods with positive temperature recordings on



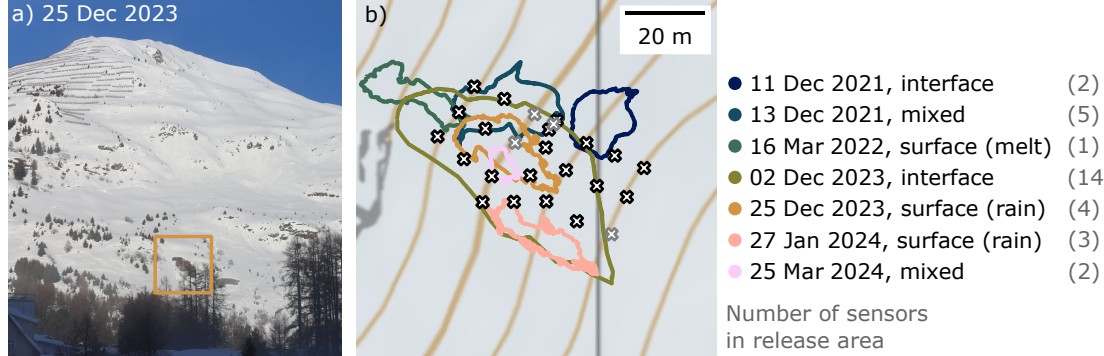

**Figure 4.** a) Picture of release area in Seewer Berg slope (rectangle) on 25 Dec 2023. b) Release areas extracted from time-lapse photographs for glide-snow avalanches on the sensor grid. Grid sensors (x) that were not used for soil LWC observations are marked in gray. The number of sensors in the release area after manual modification (without excluded gray sensors) are given in parantheses for every event. Map: Federal Office of Topography, north up, contour line spacing: 10 m.

the wedges were excluded. Such periods occurred frequently due to a generally shallow snow cover through season 2022 (Figure 2) and repeated avalanche release in season 2024. The snow LWC sensors were calibrated for snow in the laboratory (prediction uncertainty: 1.7 % (Koch, 2023)). To account for small inter-sensor or setup-induced variations, the air permittivity
was determined in-situ by the mean measurement during a dry summer time-period. When we use the term 'snow LWC' we refer to the liquid water content of the interfacial snow. We defined the interfacial snow as the lowermost 20 cm of the snowpack. For the manual snow profiles we calculated the 'snow LWC' as the mean of all measurements (capacitive probe) within the lowermost 20 cm of the snowpack. To calculate the mean we excluded measurements in dry snow (0 % LWC).

**Snow cover:** The snow cover across the Seewer Berg slope was manually classified into time periods of a continuous snow
cover and a patchy snow cover using the time-lapse photographs. A pixel in the time-lapse photographs corresponds to an area of approximately 0.25 m$^2$ (Fees et al., 2023). The snow cover was considered patchy when more than one grid sensor was snow free. Only time periods with a continuous snow cover were taken into account for measurement aggregation such as season mean values.

## 4.2 Measurement uncertainties and significance test

The measurement uncertainty of aggregated sensor measurements consists of the sensor accuracy for a single measurement and a statistical component (e.g. the standard deviation). Unless otherwise indicated, the uncertainty visualized and given in the results refers to the standard deviation. The sensor accuracies are listed in Table B1. When we compared data from two different groups (e.g. interface/surface, in/outside release area) we used the Mann-Whitney U test.





## 4.3 Spatial dependence: variogram

We determined the degree of spatial dependence between measurements across the sensor grid using the variogram. A variogram relates the variances of observations at different spatial separations (Oliver and Webster, 2015). We calculated the overall variance ($\sigma^2$) from the measurements and binned the differences between observations of our sensors uniformly by sensor distances. An exponential function was fitted to the experimental data to determine the range ($r$) (details in Appendix A). The range indicates the distance above which measurements are no longer spatially correlated (Oliver and Webster, 2015). A small

variance and a large range indicate high spatial uniformity (i.e. low spatial variability).

## 5  Results

We observed seven glide-snow avalanches on the sensor grid in the Seewer Berg slope during season 2022 and season 2024 Figure 4). In this section, we first show an overview of the winter seasons and their avalanche activity which indicated that water at the soil-snow interface was a prerequisite for avalanche release. We then analysed the source, quantity, and spatial

distribution of the interfacial water based on the recorded glide-snow avalanche events on the Seewer Berg slope.

### 5.1  Seasonal overview and interfacial water

The 2022 winter season (Figure 5a) was characterized by overall below-average new snow totals (Figure 2). The first large snowfall in December was followed by a mostly dry and warm January. At the beginning of February there was another large snowfall followed by a relatively dry February and March (Figure 5a2). The average soil LWC was $(0.35 \pm 0.01)$ m$^3$m$^{-3}$ and

the average soil temperature was $(1.5 \pm 0.5)$ °C. On Dorfberg, a total of 46 glide-snow avalanches were recorded, which is slightly below the 16-year average of 62 avalanches (Figure 2).

The 2023 winter season was characterized by below-average snowfall (Figure 2). On Dorfberg, the snow cover was sparse and intermittent and there were no glide-snow avalanches observed. This was the first season without any glide-snow avalanches recorded on Dorfberg since the start of observations in season 2009 (Figure 2).

The 2024 winter season (Figure 5b) was characterized by a first snowfall in the beginning of November followed by a large snowfall at the end of November and continuous snowfalls throughout December. This was followed by a warm and mostly dry January and February. The sum of new snow height was above average and the number of recorded glide-snow avalanches (104) was the second highest since the start of observations in season 2009 (Figure 2). Overall, the season was characterized by continuous glide-snow avalanche activity with two days of exceptionally high activity on 12 December 2023 and 25 January

2024, both after a rain-on-snow event. The average soil LWC was comparable to season 2022 with $(0.35 \pm 0.01)$ m$^3$m$^{-3}$. The average soil temperature $(3 \pm 1)$ °C was significantly higher than in season 2022 (p < 0.01, Figure 5b4,5).

The glide-snow avalanche activity in season 2022 occurred in three distinct clusters (Figure 5a1). Around the December and March clusters, we observed snow LWC of 4 % and 9 % respectively (Figure 5a6). Between the two clusters the snow LWC was generally low (< 4 %) or low air temperatures caused a quick decrease in snow LWC due to refreezing (6 Jan 2022). In







**Figure 5.** Overview (daily mean values) of a) season 2022 and b) season 2024. The background colors indicate patchy snow cover (gray), continuous snow cover (white) or rain (blue). The dashed line indicates the time of avalanche release in the Seewer Berg slope. _1) Avalanche activity on Dorfberg derived from time-lapse photographs classified as surface (orange) or interface (blue) events based on the SNOWPACK simulation. _2) Snow height simulated at the reference location (line, classified in surface/interface with SNOWPACK) and snow heights from manual snow profiles at the reference location (·) or from avalanches in the Seewer Berg slope (x). _3) Air temperature at the reference location (SNOWPACK). _4) Mean soil temperature across the sensor grid, the mean interface temperature and the interfacial snow temperature from the manual snow profiles. _5) Soil LWC recorded by the individual sensors (gray) and the mean across the grid (black). _6) Snow LWC observed in the manual snow profiles in the reference location (·) and behind the fracture line of avalanches in the Seewer Berg slope (x). The color indicates the grain type of the lowermost snow layer. _7) Snowpack bulk density colored by the dominant grain type within the snow profile. Abbreviations for grain types: PP - Precipitation particles, DF - Decomposing and fragmented precipitation particles, RG - Rounded grains, FC - Faceted crystals, MF - Melt forms, MFcr - Melt freeze crust.



comparison glide-snow avalanche activity in season 2024 occurred continuously throughout the season (Figure 5b1) and we observed overall higher snow LWC (Figure 5b6). Throughout the season we only observed one manual snow profile without snow LWC (24 Jan 2024). This snow profile was followed by a rain-on-snow event the following day. The rain reintroduced water to the soil-snow interface which we observed through water percolation into the soil (Figure 5b5). The comparison of the time of glide-snow avalanche activity and the snow LWC indicated that water at the soil-snow interface was a prerequisite

for avalanche activity in both seasons.

## 5.2 Source of interfacial water

As shown above, the observation of snow LWC in manual snow profiles at the reference location coincided with the occurrence of glide-snow avalanches on Dorfberg suggesting that interfacial water is a prerequisite for glide-snow avalanche activity. In the following, we investigate the source of interfacial water in detail based on the seven glide-snow avalanches that released on

the sensor grid in the Seewer Berg slope (Figure 4).

### 5.2.1 Interface events

We classified four glide-snow avalanches (11 Dec 2021, 13 Dec 2021, 2 Dec 2023, 25 Mar 2024) as interface events (Figure 4). The possible sources of interfacial water for interface events are melting of the basal snow layer by geothermal heat and/or capillary suction from the soil into the snow (Mitterer and Schweizer, 2012a).

All interface events released several (13, 15, 8, 2) days after snowfall on bare ground. In the continuously snow covered periods 8 days preceding the avalanche, the interface events showed constant (Figure 6a3, c3) and above-average soil temperatures across the sensor grid (Figure 6a3, c3, f3). For all avalanches, the soil temperatures below the release area were higher than in the remaining sensor grid (Figure 7a). These observations indicate that the interface events were driven by geothermal heat melting the lowermost snow layer due to high soil temperatures. In addition to the above-average soil temperatures, we

observed indications of interfacial meltwater being sucked up into and remaining in the interfacial snowpack for several days (Figure 6a4,c4).

A large snowfall (60 cm) on 24 November 2023 fully covered the wedges carrying the snow LWC sensors. The sensor at 5 cm height measured an increase in snow LWC shortly after the initial snowfall until the snow LWC reached around $(2.6 \pm 0.1)$ % (average for 26 Nov). A slower, time-shifted increase in snow LWC was observed at 10 cm which reached $(1.9 \pm 0.1)$ % and at

25 cm which reached $(1.7 \pm 0.1)$ %. The snow LWC remained elevated for several days before the sensors broke shortly before the avalanche released (02 Dec 2023) (Figure 6c4). The snow LWC sensors likely broke due to small glide or creep movements of the snowpack. However, no visible indications of glide-crack formation were observed on the time-lapse photographs. The snow LWC sensor at 5 cm peaked shortly before breaking (1 Dec). It is currently unclear if this peak was due to an increase in snow LWC, or from the deformation of the sensor shortly before breaking. The snow LWC observed in the release area, 4

days after avalanche release, was $(5 \pm 1)$ %, comparable to the snow LWC measured at the wedges before release (Figure 5b6, Figure 6c4). This increase in snow LWC before avalanche release was not observed for the other interface events due to shallow snow heights which did not sufficiently cover the snow LWC sensors. However, the manual snow profiles in the release areas



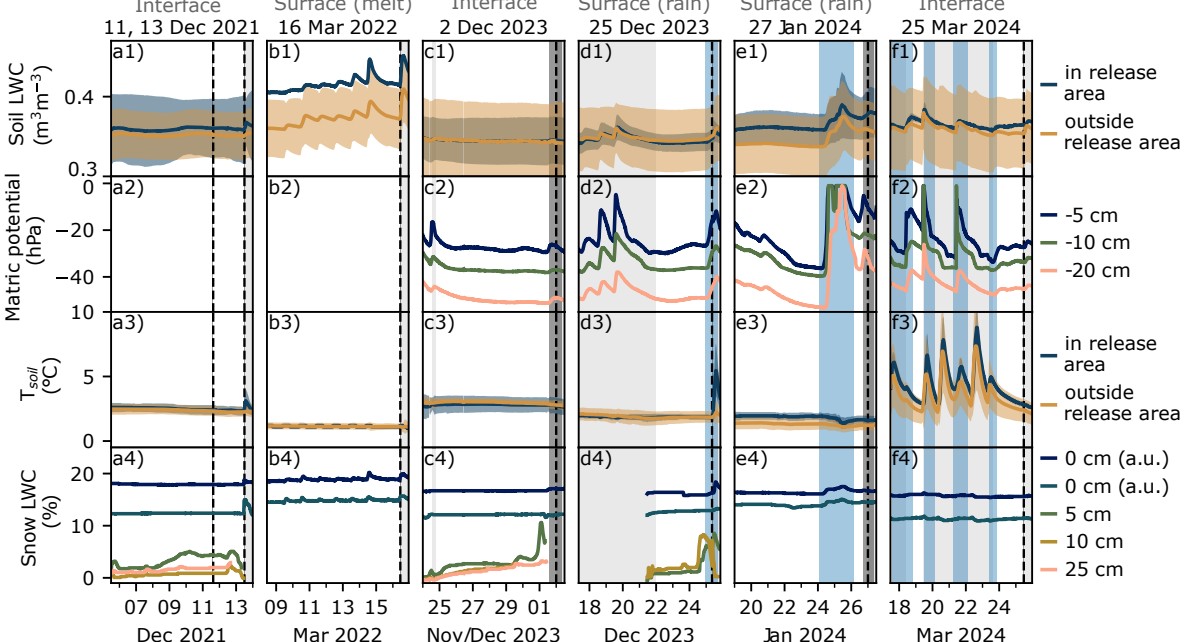

**Figure 6.** Soil and snow properties before glide-snow avalanche release: events are separated in columns (a-f) and the rows (1-4) show the measurements starting 8 days before avalanche release. The background color indicates a patchy (gray), continuous (white) snow cover, or rain (blue). The dashed line indicates the time of avalanche release in the Seewer Berg slope with the range of uncertainty in dark gray. The parameters include: _1) the soil LWC separated between sensors that were located in the avalanche release area (orange) and the remaining sensors in the grid (blue). _2) The soil matric potential. _3) The soil temperature shown separately for the sensors that were located in the avalanche release area and the remaining sensors. _4) The snow LWC sensors at three heights and interface sensors (in arbitrary units).

showed melt forms in the lowermost snow layer for all interface events independent of the majority grain type in the remaining profile (Figure 5a6, a7, b6, b7). This indicates that water existed in the lowermost snowpack layer before avalanche release.

Capillary suction of water from the soil into the snow has been suggested as another potential source of interfacial water (Mitterer and Schweizer, 2012b). We observed for two events (11 Dec 2021, 2 Dec 2023) that the soil LWC leading up to glide-snow avalanche release was constant and its quantity was comparable between the release area, the remaining sensor grid and the season mean (Figure 6a1,c1,f1, Figure 7b). Based on the measured saturation, capillary suction was unlikely to contribute substantial amounts of interfacial water across the sensor field (Lombardo et al., 2024).

For the other two interface events (13 Dec 2021, 25 Mar 2024) we observed above-average soil LWC below the release area (13 Dec 2021) and across the entire grid (25 Mar 2024). The 13 Dec 2021 avalanche released shortly after the 11 Dec 2021 avalanche which was classified as an interface event. However, on the day before avalanche release (12 Dec 2021) positive air temperatures occurred in combination with the shallow snowpack (27 cm, 16 Dec) which could have caused surface meltwater





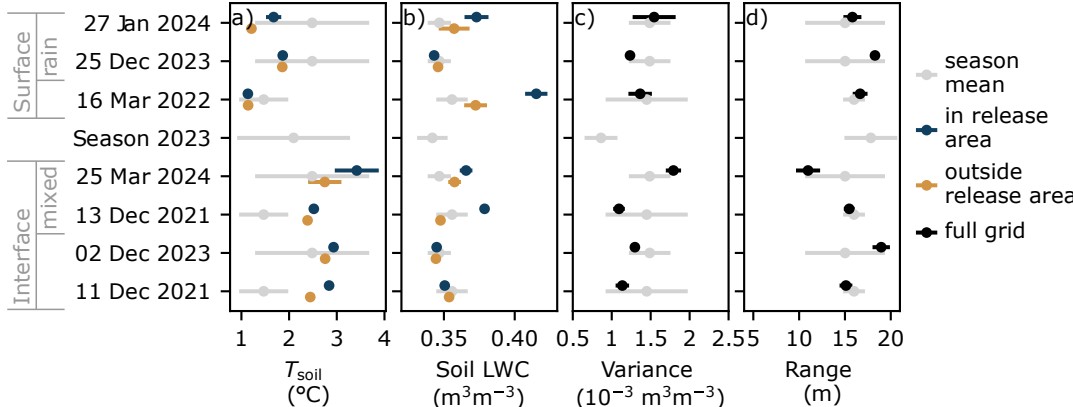

**Figure 7.** Comparison of average soil properties before avalanche release with the season mean (gray) sorted by the source of interfacial water. The a) soil temperature and b) soil LWC were separated in sensors in (blue) and outside (yellow) the release area. The spatial distribution of the soil LWC was quantified across the entire grid (black) using c) the soil LWC variance and d) the soil LWC range (Appendix A). The average for the avalanche events was calculated for the snow-covered time-periods 8 days before avalanche release. For rain-on-snow events the average was calculated from the start of the rain to the time of avalanche release.

formation and percolation (Figure 5a2, a3, a5). We therefore classified this avalanche as a mixed event, i.e. a combination of

interface and surface event.

The 25 Mar 2024 avalanche differed from the other interface events as it occurred in spring after a snowfall on snow-free ground. Several rain events on bare ground in the 8 days leading up to the release caused above-average soil LWC (Figure 6f1). However, soil LWC, matric potential and interface sensors showed no indication of meltwater percolation (Figure 6f1, f3, f6) after the snowfall occurred. Due to the high soil temperatures, we suspect that the main source of interfacial water was due

to geothermal heat, yet due to the positive air temperatures and shallow snowpack, contributions from meltwater percolation are also possible (Figure 5b3). The snowpack across the Seewer Berg slope melted out within the day of avalanche release which prevented us from recording a manual snow profile in the release area. This event was also classified as a mixed event. The long-term observations of avalanches on Dorfberg suggest that such mixed events occur frequently. A majority (85%) of glide-snow avalanches that released within the first 8 days after the first major snowfall of the season showed surface meltwater

formation in the SNOWPACK simulations. These events are likely mixed events where water originates from geothermal heat and surface melt (Figure C1).

### 5.2.2 Surface events

We classified three avalanches as surface events. These include one event due to meltwater percolation (16 Mar 2022) and two events due to rain (25 Dec 2023, 27 Jan 2024).



For the meltwater-driven glide-snow avalanche (16 Mar 2022), we observed diurnal peaks of meltwater infiltration into the soil in the 7 days preceding avalanche release. These peaks occurred around 13:00 local time (LT), coinciding with the expected time for diurnal meltwater percolation across the soil-snow interface (Figure 6b1, b4). The vertical soil sensor profiles showed that the meltwater percolated through the soil (Figure 8a). The diurnal observation of water infiltration in the soil indicates that water reached the soil-snow interface on several days before the avalanche released. The soil LWC across the entire slope was substantially above the season mean, and the soil LWC in the release area was significantly higher than the soil LWC in the

remaining slope (p<0.01, Figure 7b).

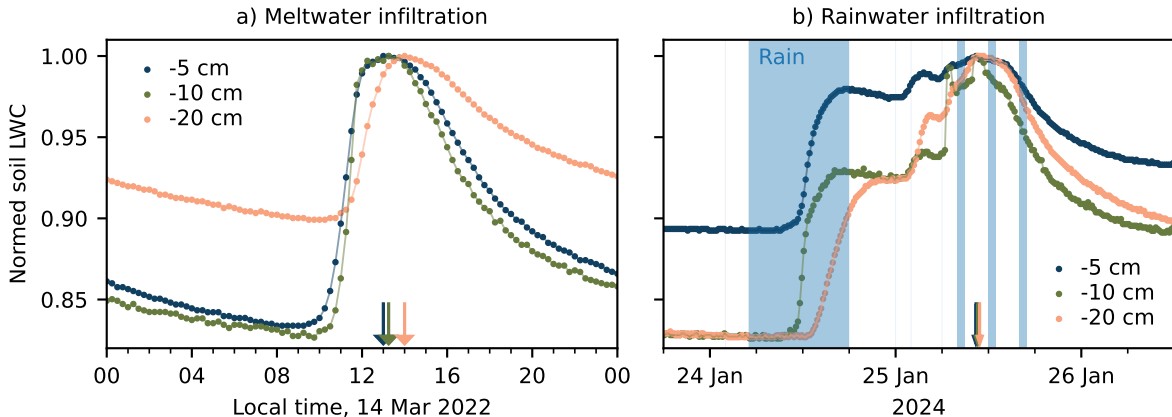

**Figure 8.** Normalized LWC for three soil depths which show a) meltwater and b) rainwater infiltration. The soil LWC was normalized with the maximum value to better visualize the time-shift in the percolation. The arrows indicate the time of the maximum. The time of rain (blue) was extracted from the SNOWPACK simulation at the reference location.

For both rain-on-snow events (25 Dec 2023, 27 Jan 2024) we observed that the rain percolated through the snowpack into the soil. The soil LWC, matric potential and interface sensors all showed an increase in available water at the interface and

in the soil (Figure 6d1, d2, d4, e1, e2, e4). For the 27 Jan event the cold percolating water (Figure 8b) also caused the soil temperature to decrease for a short time (Figure 6e3) and the matric potential sensors showed that soil saturation (-1 hPa) was reached temporarily (Figure 6e2). Both avalanches released after the initial observation of water percolation into the soil and one day (25 Dec 2023) and two days (27 Jan 2024) after the rain. This indicates that water reached the soil-snow interface for around half a day to one and a half days before the avalanche released.

The soil LWC for the rain event of 27 Jan 2024 was significantly higher in the release area compared to the rest of the grid, a pattern we did not observe for the 25 Dec 2023 avalanche (Figure 7b). It has to be noted that the 25 Dec avalanche occurred 4 days after the previously snow-free Seewer Berg slope was covered again by snow. As a result, the snowpack was likely comparable to an early season (interface event) snowpack with a low bulk density (Figure 5a7, b7) before the rain occurred. We suspect that due to the early season snowpack, smaller quantities of interfacial snow liquid water content were sufficient



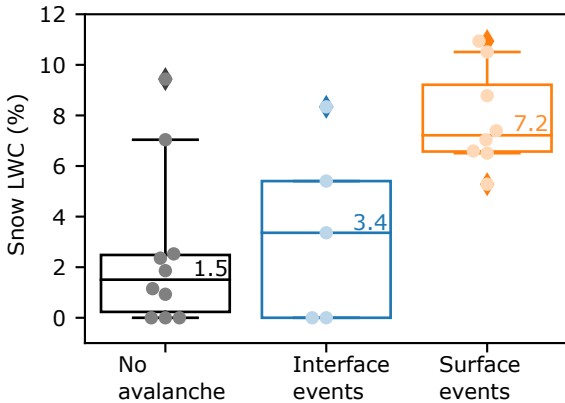

**Figure 9.** Snow LWC observed in the manual snow profile (reference location) grouped by the activity (no avalanche (n=10), surface event (n=8), interface event (n=5)) which occurred ± 4 days of the manual snow profile. The suspected source of interfacial water (surface/interface) was classified based on the SNOWPACK simulation. The median value is given and the whiskers indicate the 5th and 95th percentile. Surface events showed significantly higher snow LWC than interface events (p = 0.04). Snow profiles were filtered manually to exclude the end of the season and a lack in continuous snow cover.

for avalanche release (Figure 9). This could explain why we observed an onset of water percolation into the soil but not yet above average soil LWC.

Geothermal heat or capillary suction likely did not contribute to the formation of interfacial water for the observed surface events. For all surface events, we observed that the soil temperatures across the grid were below average, reducing the potential water contribution through geothermal heat (Figure 7a). For the 25 Dec 2023 rain event, soil LWC was not close enough to

saturation to allow for capillary suction (Figure 7b). For the 27 Jan 2024 rain event, the matric potential sensors indicated that the soil was close to saturation (-1 hPa, Figure 6e2), which would technically allow for capillary suction from the soil into the snow (Lombardo et al., 2023, 2024). However, the high saturation was only reached when the water from the soil-snow interface percolated into the soil, so the water was already present at the soil-snow interface.

### 5.3  Quantity of interfacial water

The snow LWC that we observed for interface events was (3 ± 1) % three days before release (snow LWC sensor, 02 Dec 2023), (5 ±1) % (6 Dec 2023), and (3± 2) % (16 Dec 2021) in the release area.

The snow LWC that we observed for surface events was (9 ± 2) % (surface-melt, 16 Mar 2022), (9 ± 1) % (29 Dec 2023), and (9 ± 3) % (30 Jan 2024), measured at the reference location. The release area profile likely underestimated snow LWC due to cold air temperatures during the night before the profile was recorded. Overall, the observed snow LWC in the reference

profile around the time of avalanche release was significantly higher than for the interface events (p = 0.04, Figure 9).





## 5.4 Spatio-temporal distribution of interfacial water

We analyzed the spatio-temporal distribution of soil LWC to assess its variability for interface and surface events. In general, based on a geostatistical analysis, it can be expected that spatio-temporal variability decreases with increasing range and/or

decreasing variance. A low spatial variability indicates a more uniform distribution of soil LWC across the slope. We excluded the avalanche of 13 Dec 2021 from this analysis because the avalanche two days prior (11 Dec 2021) caused a patchy snow cover and the grid sensors subsequently experienced variable conditions.

Across all seasons, the observed average range during snow-covered periods was $(17 \pm 1)$ m and the variance was $(1.2 \pm 0.3) \times 10^{-3} \text{m}^3\text{m}^{-3}$. The average values were comparable between seasons (Figure 7c,d). The observed glide-snow

avalanches showed an average range of $(16 \pm 3)$ m and an average variance of $(1.3 \pm 0.2) \times 10^{-3} \text{m}^3\text{m}^{-3}$ at the time of release. The spatio-temporal development of the soil LWC variability before avalanche release showed three distinct spatio-temporal behaviours.

The first behaviour was observed for one interface event (11 Dec 2021), one mixed event (25 Mar 2024), and one rain-on-snow event (25 Dec 2023). For these events, the time of avalanche release coincided with a local minimum in soil LWC

variability (low variance and/or large range, Figure 10a,c,e). The variability decreased throughout the day(s) prior to release and we observed fluctuations of increasing/decreasing range at similar variances before release. For these events no water infiltration occurred from the interface into the soil. It is currently unclear what drives the spatio-temporal evolution of the soil LWC variability without water infiltration from the snowpack.

Another version of the first behaviour was observed for an interface event (02 Dec 2023). We observed increases and

decreases in range, but no substantial decreases in variance (Figure 10b). However, the overall soil LWC variability observed during the 7 days before release was comparable with the soil LWC variability of the surface events on 16 Mar 2022, 25 Dec 2023, and 27 Jan 2024 (Figure 10c,d,f). The avalanche finally released during a heavy snowfall with rapidly increasing snow load. The increasing snow load may have contributed to the critical conditions resulting in glide-snow avalanche release.

The second behaviour was observed for the surface event driven by melt (16 Mar 2022, Figure 10d). The recurring diurnal

meltwater percolation into the soil resulted in a repeating pattern of decreasing and increasing spatial variability. The avalanche released when the overall soil variability decreased as part of the diurnal pattern. However, the avalanche did not release at the minimum soil variability, which had been reached prior to release. It is currently unclear if a change in conditions for example of the snowpack, was necessary to cause avalanche release.

The third behaviour was observed for the rain-on-snow event of 27 Jan 2024 (Figure 10f). The soil LWC uniformity 7 days

before avalanche release was quantitatively comparable to the uniformity at the time of release. However, 7 days before release there was no interfacial water available (0 % snow LWC, 24 Jan 2024, Figure 5b6). The rain percolated to the soil-snow interface and into the soil, but also initially increased the soil LWC variability. We suspect that this increase in soil LWC variability was due to the rain water percolation along preferential flow paths through the snowpack that was not isothermal yet. This would introduce a heterogeneous pattern of water percolation into the soil and increase variability. The avalanche

released once the soil LWC variability had decreased again substantially within the next two days.



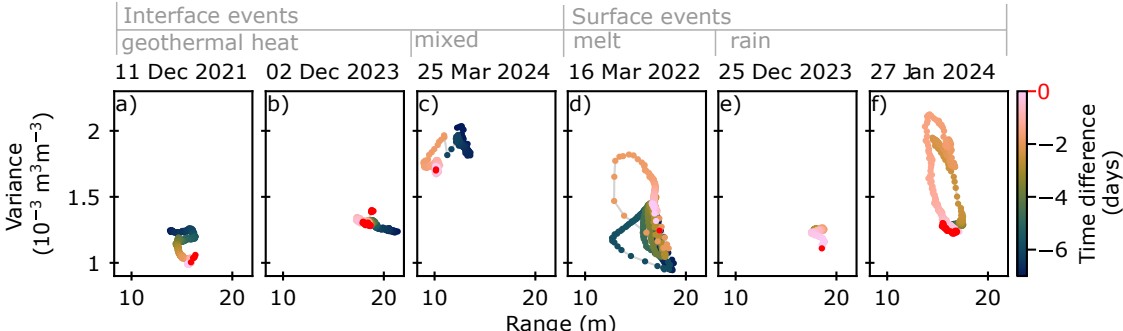

**Figure 10.** Temporal evolution of the spatial variability before avalanche release (red, t = 0). The spatial variability decreases with decreasing soil LWC variance and/or increasing soil LWC range. A low spatial variability indicates a more uniform soil LWC distribution. More than one red point indicate the uncertainty in time of release due to limited visibility in the time-lapse photographs.

## 6 Discussion

We observed snow and soil conditions in space and time for seven glide-snow avalanches before and during avalanche release. This allowed us to investigate the source, quantity, and spatial distribution of the interfacial water involved in glide-snow avalanche release. The small number of observed glide-snow avalanches currently limits the generalization of findings. How-
ever, when possible we supported our findings with the 16-year dataset of Dorfberg (Fees et al., 2023).

### 6.1 Source and quantity of liquid water

We observed a wide range of sources for interfacial water for glide-snow avalanches including geothermal heat (n=2), meltwater (n=1), rain (n=2), and mixed types with geothermal heat and melt (n=2). In contrast to Ceaglio et al. (2017), we did not observe indications of capillary suction of water from the soil into the snow due to low soil saturation (matric potential) (Lombardo
et al., 2023). Sufficient soil saturation for capillary suction was mostly observed when water originated from melt or rain and percolated into the soil (for detailed investigation od capillary suction see Lombardo et al. (2024)).

We used a combination of meteorological, snow, and soil observations to classify the avalanches based on their suspected source of interfacial water. These proxies were needed due to the lack of available measurement methods that allowed for spatio-temporal snow LWC monitoring. Locally it was possible to measure the snow LWC using a capacitive LWC sensor
mounted on a wedge. This sensor measured one instance where we suspect geothermal heat melted the new snow at the soil-snow interface and the available water was sucked into the snow and remained there for several days before avalanche release (Figure 6,c4). This is our only observation of this process because the wedge (height: 30 cm, Figure 3d) introduced preferential melt in its proximity if it was not sufficiently covered by snow.

Another method to measure the snow LWC is through snow profiles. Using snow profiles, we found non-zero snow LWC
during times of avalanche activity, in line with observations by Fromm et al. (2018) and Ceaglio et al. (2017). In addition, we observed that snow LWC was significantly lower for interface events than for surface events (Figure 9), in line with Maggioni





et al. (2019). This observation may be related to the snow cover below the release area, called the stauchwall. It is suspected that interface events driven by geothermal heat are common when the relatively warm soil is newly covered in snow. This new snow has a lower density than the snow cover in early spring (Figure 5a,b7). Bartelt et al. (2012) suggested that lower snow densities are associated with a weaker stauchwall. As a result, less interfacial water may be needed for a glide-snow avalanche to release.

We observed for two out of four interface events, that interfacial water may have originated from meltwater percolation in addition to geothermal heat. Our long-term observations of avalanche activity with time-lapse photography and SNOWPACK simulations support this finding (Figure C1). These common mixed events (interface and surface events) could explain why Dreier et al. (2016) found warmer air and snow surface temperatures for avalanches that released in early winter (before mid February).

## 6.2 Spatial variability

Previous studies (Fromm et al., 2018; Ceaglio et al., 2017; Maggioni et al., 2019) found that the soil temperature and the soil LWC were related to snow gliding. Generally, we observed higher local soil temperatures or soil LWC (or both) below the release areas compared to the rest of the slope. This suggests that the local temperature/soil LWC may indicate the location of glide-snow avalanche release. These local differences highlight the importance of spatio-temporal monitoring. Further investigation focusing on the cause of local differences such as soil inhomogeneities and preferential flow patterns is necessary to narrow down the cause and timing for glide-snow avalanche release at the slope scale.

The investigation of the soil LWC across the grid showed that spatial variability often decreased before avalanche release (Figure 10). This supports the hypothesis from recent pseudo-3D modelling (Fees et al., 2024) that the spatial variability of the basal friction is important for glide-snow avalanche release. The range across the Seewer Berg slope varied at the meter length scale within the days to hours before avalanche release. When interpreting the soil LWC variability, interface and surface events have to be separated. For surface events, we observed water percolation from the interface into the soil. In this case, soil LWC uniformity may be a good proxy for the interfacial water distribution. The percolation may indicate locations with more interfacial water and thus reflect the spatial variability of water at the soil-snow interface. For interface events, we did not observe water percolation from the interface into the soil. Instead, we observed the capillary rise of water (formed at the soil-snow interface through geothermal heat) into the snow (Figure 6c4). The relationship between the soil LWC variability and the interfacial water is not yet clear and requires further investigation.

The spatial distribution of soil temperature may be a suitable proxy for the interfacial water distribution in interface events. However, the soil temperature did not fulfil the assumption of a random field necessary to assess spatial variability using the variogram. While the 8 m x 8 m sensor grid spacing was suitable for an initial investigation of spatial variability, smaller distances between sensors would provide more detailed spatial information. More available measurements would also allow for the investigation of potential directional anisotropies across the slope.

Overall our results indicate that a sufficient quantity (snow LWC interface events: ∼3 %, surface events: ∼7 %) and low spatial variability of water are needed for glide-snow avalanche release. During seasons with continuous glide-snow avalanche



activity (e.g. season 2024) these potentially critical conditions prevailed throughout most of the season. When those potentially critical conditions exist, snow loading with new snow may then facilitate avalanche release (e.g. 2 Dec 2024). Dreier et al. (2016) also found that avalanches often released after snowfalls based on observations of season 2012 on Dorfberg, which was characterized by continuous glide activity (Figure 2) and likely prevailing critical conditions.

Our classification into interface and surface events based on SNOWPACK simulations generally agreed well with our field observations. However, to quantify the snow LWC leading up to interface events, the soil has to be implemented (ideally driven with measured soil LWC and temperature) in the simulations. To investigate, predict, or monitor avalanches at the slope scale, spatio-temporal monitoring of soil LWC, temperature, and, if possible, snow LWC seems promising and necessary.

## 7   Conclusions and Outlook

We installed a spatio-temporal soil and local snow monitoring setup in an avalanche-prone slope. During season 2022 to 2024 we observed seven glide-snow avalanche releases on our sensor grid. We analyzed these avalanches in detail to investigate the source, quantity and spatial distribution of liquid water before and during avalanche release. The interfacial water we observed originated from (i) geothermal heat (n=2), (ii) meltwater percolation from the surface (n = 1), (iii) rain (n=2) and (iv) a combination of geothermal heat and meltwater percolation (n=2). Our results show that for glide-snow avalanche release

a sufficient amount of snow liquid water and its spatial distribution are important. For interface (geothermal heat) events we observed lower snow LWC (∼3 %) before/after avalanche release than for surface (melt, rain-on-snow) events (∼7 %). For most events, the release area showed locally (i) higher soil temperatures during the 8 days preceding an avalanche associated with geothermal heat and (ii) higher soil LWC below the release area during the 8 days preceding a surface event. The spatial variability of soil LWC repeatedly (4/6 avalanches) showed a local minimum at the time of release. In the future, with continued

observation, the spatio-temporal investigation of the soil will help to quantify the drivers for glide-snow avalanche release depending on the source of liquid water and at the slope scale. Linking the quantity and spatial distribution of interfacial water to its drivers will be an important step to more accurately predict the time of avalanche release. In addition to improved process understanding continued spatio-temporal monitoring is a promising approach to narrow down length and time scales as well as suitable proxies for glide-snow avalanche monitoring that could be used for mitigation or forecasting.

## Appendix A:  Variogram


We determined the degree of spatial dependence between measurements using the variogram. A variogram relates variances of observations at different spatial separations (Oliver and Webster, 2015). We calculated the overall variance ($\sigma^2$) from the measurements and binned the distances of our sensors uniformly. We fitted an exponential function to our observations

$$\gamma = \sigma^2 \exp(-\frac{x}{a}) \tag{A1}$$

which does not take into account a nugget effect. As the exponential function asymptotically approaches the variance ($\sigma^2$) the range was defined as $r = 3a$ where 95 % of the variance was exceeded (Oliver and Webster, 2015).



## Appendix B: Sensor specifications

Sensor specifications are listed in Table B1.

**Table B1.** Sensor specifications during typical winter conditions as provided by the manufacturer. We listed the worst case accuracy.

| Sensor | Manufacturer/Source | Parameter | Range | Accuracy |
|---|---|---|---|---|
| Tensiomark | ecoTech Umwelt-Messsysteme GmbH (2014) | Matric potential | $1 - 10^7$ hPa | $\pm 30$ hPa |
| | | Temperature | -40 to +80 °C | $\pm$ 0.1 °C |
| TEROS11 | Meter Group (2024) | LWC | 0.00–0.70 $m^3 m^{-3}$ | $\pm 0.03$ $m^3 m^{-3}$ |
| | | Temperature | -40 to +60 °C | $\pm 0.5$ °C |
| EC5 | Meter Group (2023) | LWC | $0 - 1$ $m^3 m^{-3}$ | $\pm 0.03$ $m^3 m^{-3}$ |
| T107 | Campbell Scientific Inc. (2018) | Temperature | -35° to +50 °C | $\pm 0.4$ °C |

## Appendix C: Source of water in avlanches after first snowfall

We investigated the source of interfacial water for glide-snow avalanches that released within 20 days after the first snow fall of the season on bare ground. The first season snowfall was defined as the snowfall that starts the seasons snowpack and does not melt within a few days. This investigation was based on the the long-term Dorfberg observations (2009-2024) which consist of glide-snow avalanche activity extracted from time-lapse photographs and SNOWPACK simulations (Fees et al., 2023). The avalanches were classified in interface/surface events using the SNOWPACK simulations. The cumulative
avalanche release probability increased substantially within the 8 days after the first snowfall (85 %, Figure C1) and for most avalanches meltwater was a potential source of interfacial water (83 %, surface events). These events are likely mixed events were contributions from geothermal heat and meltwater formation contributed interfacial water.





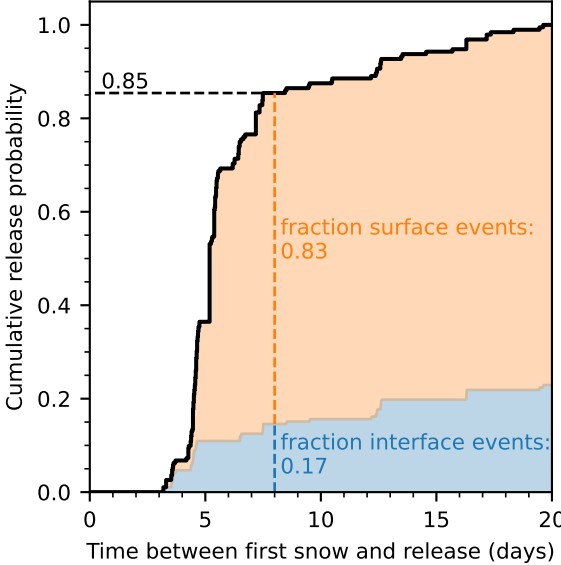

**Figure C1.** Cumulative release probability for 20 days after the first snowfall on snow-free ground. The avalanches were classified in surface (orange) and interface (blue) events.

*Author contributions.* Conceptualization: all authors, Methodology: all authors, Formal analysis: A.F., Investigation: A.F., A.H., M.L., Data Curation: A.F., Writing - original draft: A.F., Writing - review & editing: all authors, Visualization: A.F., Supervision: A.H., J.S., Funding
acquisition: P.L., J.S.

*Competing interests.* Jürg Schweizer is a member of the editorial board of The Cryosphere.

*Acknowledgements.* This research was supported by the Swiss National Science Foundation (grant no. 200021-212949). Colormaps by Crameri et al. (2020). The authors would like to thank Katrin Meusburger for help with installing the first sensors and Grégoire Bobillier for helpful discussions. We would also like to thank the electronics group at SLF (Chasper Buchli) for help with the field setup and Flavia
Mäder, Moritz Altenbach, and Leonardo Stapelfeldt for help with field work.

*Data availability.* All sensor measurements will be made available on Envidat before publication.



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
