# Peer review of "The source, quantity, and spatial distribution of interfacial water during glide-snow avalanche release: experimental evidence from field monitoring"

_EGUsphere, 2024_

## Author Comment (AC1)

**Reply to Referee 1:**

We would like to thank Referee 1 for reviewing our manuscript and for their helpful feedback. We addressed the referee's comments in blue color.

The paper investigates the source, quantity, and spatial distribution of interfacial water before and during glide-snow avalanche release. As the liquid water at the snow soil interface is a key predisposing factor for glide avalanche release, this paper significantly contributes to the comprehension of this natural phenomenon. I really appreciate the efforts to observe and measure the snow and soil properties jointly, since together they represent a highly dynamic and connected porous medium.
The paper is well written, minor issues are listed below:

Line 12: maybe better "for the majority of the snow avalanches considered in the study" rather than four avalanches

We will implement this change.

Line 14: please specify what kind of soil properties have to be considered

We will specify that the soil properties refer to the soil liquid water content (LWC) and temperature.

Line 17: change infrastructure into infrastructures

We will keep the word infrastructure (according to Merriam Webster: https://www.merriam-webster.com/dictionary/infrastructure).

Line 29: I would add layer after liquid water

We will implement this change.

Line 51: I would specify at what soil depth grid sensors measured the soil temperature and LWC across the slope

We will implement this change.

Caption in Figure 2: Please explain the meaning of the different colours in the graph

We will add the colors to the caption.

Line 75: How the slope is protected? Because of the lower slope angle? Because of some active protections?

The slope is protected from glide-snow avalanches due to the local topography consisting of a small hill above the slope. This protects the reference site from the larger avalanche prone-slopes at higher elevations. We will explain this in the revised manuscript.

Line 76: May you provide some more information about the soil? What is the soil classification? Moreover I think that not only the texture but also the soil organic matter content could influence the water retention.

The soil was classified as a sandy loam and soil organic matter content was not measured. The water retention of the soil in the Seewer Berg slope and its potential connection to the capillary suction of water from the soil into snow was investigated in detail in Lombardo et al. (2025) (accepted for publication). In Lombardo et al. (2025) we also provide a data repository (Envidat) including the details of the soil profile. We will refer to Lombardo et al. (2025) and the data repository in the revised manuscript.

Line 308: I think that in Ceaglio et al. 2017 the potential contribution of groundwater in the study area could have been higher than in your study site, saturating the soil with a potential higher movement of water from soil to the lowermost layer of the snowpack.

Thank you for pointing this out, we will mention the groundwater source in Ceaglio et al. (2017).

Line 337: I fully agree that soil inhomogeneities and preferential flow patterns could explain the spatial variability of the local temperature/soil LWC. I would add also the plant cover inhomogeneities which could change the surface roughness.

We will mention the plant cover inhomogeneities as a potential contribution.

Line 389: correct avalanches

We will correct this typo.

References:
Lombardo, M., Fees, A., Udke, A., Meusburger, K., van Herwijnen, A., Schweizer, J., and Lehmann, P.: Capillary suction across the soil-snow interface as a mechanism for the formation of wet basal layers under gliding snowpacks, Journal of Glaciology, accepted for publication, 2025

---

## Author Comment (AC2)

**Reply to Referee 2**

We would like to thank Referee 2 for their review and feedback on our manuscript. We addressed the referee's comments in blue color.

Although the investigations are based on only seven events the manuscript is an important contribution to better understand the influencing factors of glide snow avalanches.

However, there are major deficiencies with respect to diagrams and figures; in some diagrams the various graphs cannot really be distinguished (since the colors are very similar); some other figures are too small to discern specific details – for more information see below.

Specific points:
Line 19: it should be read as `Fankhauser´ [not `Frankhauser´]; the authors should correct the name also in the References.

Thank you for pointing this out, we will correct the citation.

Table 2: in the second line of Table 2 it should read as `(24,2,2)´ [not `(20, 2,2)´].

We will correct this typo.

Figure 4:
1) it is difficult to distinguish between the various colors in Figure 4b; please use other signatures.

   The contrast between the colors in Figure 4b is not optimal. However, we prioritize the use of colors from a colorblind-friendly colormap which reduces contrast between single colors. We find that using dotted or dashed lines to distinguish the avalanche outlines does not improve readability. For this reason, after several alternative trials, we decided to keep the original colors and solid outlines.

2) It would be useful to give a short note on the criteria of interface, mixed and surface events; certainly, there is a description in the paper, but it would be nice to have that information also in the caption of the diagram.

   We will add a short description of interface, surface, and mixed events to the caption.

Line 187: does the wording `below the release area´ mean `in the release area´? (with reference to the following figures).

Yes, and thanks for pointing this out. The wording 'below the release area' refers to 'in the release area'. We will replace all instances of 'below the release area' with 'in the release area' throughout the revised manuscript.

Figure 6:
1) it is difficult to distinguish between the various colors in Figure 6.

   We agree that the contrast between colors is not optimal. However, we chose the colors as a tradeoff between contrast and being colorblind-friendly. Colors from another colorblind-friendly colormap (e.g. viridis in python) did not improve overall contrast. We will keep the original colors.

2) the authors should explain the horizontal shades (blue, grey, orange...) in the first row (soil LWC) and third row (Tsoil).

   We will add in the caption that the horizontal shades indicate the standard deviation from the sensors in/outside the release area.

3) several lines are not visible (e.g. in c4 it is not possible to discern the line for Snow LWC in 10cm – it is visible only if one zooms into the figure).

   To improve readability, we will move the line for snow LWC (-10 cm) to the foreground of the graph. We agree that some lines (e.g. in row _3) overlap. However, we prioritize consistency in y-axis scaling between columns for easier comparability over resolving small differences within the measurements. All data will be provided in the Envidat data repository which facilitates more detailed investigations at for example smaller time steps when needed.

[Figure]

Figure 1: New version of Figure 6c4 where we moved the measurement at 10 cm to the foreground to improve readability. We also indicated the increase in snow LWC (red arrow) which refers to the referee's comment on lines 239 – 240 below.

Line 210: the authors write: `...we observed above-average soil LWC below the release area´

1) there is no specification on the average of soil LWC.

The average of soil LWC refers to the average across the winter season, as shown in Figure 7 in the manuscript. We will specify this in the text and point out that we define the winter season as the snow-covered time-period from 15 Oct to 15 Apr.

2) does the wording `below the release area´ mean `in the release area´?
Yes, see answer above.

Figure 7: it is difficult to distinguish between blue and black dots.

We will change the full black circles to open black circles to increase readability.

[Figure]

Figure 2: New version of Figure 7 with different markers of the full grid.

Line 217: the authors write: ` ...up to the release area caused above-average soil LWC ´. There is no specification on the average of soil LWC.

To clarify 'above average' we will specify that this refers to the mean soil LWC across all sensors and the entire winter season. In addition, we will refer to Figure 7 which visualizes the season mean values for soil LWC. We will also add in the caption of Figure 7 that the winter season was defined as the snow-covered time-period between 15 Oct to 15 Apr.

Line 218: the authors write: ` ... no indication of meltwater percolation (Figure 6f1, f3, f6) ... ´. Where is diagram f6?

This is a typo, we will change f6 to f4.

Line 239 - 240: the authors write: `The soil LWC, matric potential and interface sensors all showed an increase in available water at the interface and in the soil (Figure 6d1, d2, d4, e1, e2, e4) ´. The described increase cannot be seen in Figure 6e4.

There is a small increase in snow LWC (Figure 6e4) in both sensors at 0 cm, which we indicated in Figure 1 with a red arrow. The increase is less pronounced than in the soil LWC (Figure 1e1) and the matric potential (Figure 1e2). However, we prioritize the constant scaling of the graphs throughout all events visualized in the Figure for easier readability and comparability.

Line 285 – 287: The sentence `However, the overall soil LWC.... of the surface events on 16 Mar 2022, 25 Dec 2023, and 27 Jan 2024 (Figure 10c, d, f) ´ is not clear. The authors should improve that sentence.

We will remove this sentence to focus on the spatio-temporal dynamics instead of the quantitative values.

Line 292: the authors write: ` ...minimum soil variability... ´. I think the authors mean ` ...minimum soil LWC variability... ´

We will correct this typo.

Figure 10: the authors write in the caption: `Temporal evolution of the spatial variability before avalanche release... ´. I think the authors mean `Temporal evolution of the spatial soil LWC variability before avalanche release ... ´

We will correct this.

Line 330: The term `warmer air temperature´ is not reasonable; it should mean `higher air temperature´

We will correct this.

Line 334 - 335: the authors write: `Generally, we observed... below the release areas compared to the rest of the slope´ Does the wording `below the release area´ mean `in the release area´?

Yes, we will adjust this. See answer above.

Line 357: I think there must be a mistake when the authors indicate an avalanche release on 2 Dec 2024. (the paper was submitted before that date).

Correct, we will change to 2 Dec 2023.